# Combined Low Dose of Ketamine and Social Isolation: A Possible Model of Induced Chronic Schizophrenia-Like Symptoms in Male Albino Rats

**DOI:** 10.3390/brainsci11070917

**Published:** 2021-07-11

**Authors:** Suzanne Estaphan, Alexandrina-Stefania Curpăn, Dalia Khalifa, Laila Rashed, Andrei Ciobica, Adrian Cantemir, Alin Ciobica, Constantin Trus, Mahmoud Ali, Asmaa ShamsEldeen

**Affiliations:** 1Physiology Department, Faculty of Medicine, Cairo University, Cairo 12613, Egypt; sestaphan@kasralainy.edu.eg (S.E.); asmaa82shamseldeen@gmail.com (A.S.); 2ANU Medical School, Australian National University, Canberra 2605, Australia; 3Department of Biology, Faculty of Biology, “Alexandru Ioan Cuza” University of Iasi, Carol I, 700506 Iasi, Romania; andracurpan@yahoo.com (A.-S.C.); alin.ciobica@uaic.ro (A.C.); 4Psychiatry Department, Faculty of Medicine, Cairo University, Cairo 12613, Egypt; daliakhalifa_z@yahoo.com; 5Biochemistry Department, Faculty of Medicine, Cairo University, Cairo 12613, Egypt; lailarashed@kasralainy.edu.eg; 6Department of Pschiatry , “Grigore T. Popa” University of Medicine and Pharmacy, 16, Universitatii Street, 700115 Iasi, Romania; thereau_86@yahoo.com; 7Department of Morphological and Functional Sciences, Faculty of Medicine, Dunarea de Jos University, 800008 Galati, Romania; 8Biotechnology Program, Faculty of Agriculture, Cairo University, Giza, Cairo 12613, Egypt; mahmoud.bioinfo@gmail.com

**Keywords:** schizophrenia, rat, ketamine

## Abstract

While animal models for schizophrenia, ranging from pharmacological models to lesions and genetic models, are available, they usually mimic only the positive symptoms of this disorder. Identifying a feasible model of chronic schizophrenia would be valuable for studying the possible underlying mechanism and to investigate emerging treatments. Our hypothesis starts from the observation that combining ketamine with isolation could result in long-lasting neuro-psychological deficits and schizophrenia-like features; thus, it could probably be used as the first model of chronic schizophrenia that emphasizes the characteristic of having a multifactorial etiology. By the means of this study, we investigated the effects of ketamine administration combined with isolation in inducing schizophrenia-like symptoms in male albino rats and the brain reactive oxygen species levels. Our results showed that the number of lines crossings in the open field test, the number of open arm entries in the elevated plus maze, and the spontaneous alternations percentage in the Y-maze were significantly lower in the ketamine + isolation group compared to both the control and ketamine + social housing group (*p* < 0.05). Furthermore, the ketamine + isolation intervention significantly increased the MDA levels and decreased the GPx levels both in the hippocampus and the cortex of the rats. In addition, our premise of creating a model capable of exhibiting both positive and negative symptoms of schizophrenia was also based on adding the aripiprazole treatment to a group of rats. Therefore, we compared the ketamine + social isolation group with the ketamine + social isolation + aripiprazole group in order to attempt to discover if the antipsychotic drug would significantly decrease the potential positive schizophrenia-like symptoms induced by social isolation and ketamine. Given that we obtained significant results, we cautiously presume that this might be an important step in developing our animal model capable of illustrating both positive and negative symptoms of schizophrenia. This study could be a first step towards the creation of a complex animal model capable of exhibiting the multifactorial origin and manifestation of schizophrenia.

## 1. Introduction

Schizophrenia is a multifactorial disease with typically late adolescence or early adulthood onset. By some authors, it is considered to be a neurodevelopmental disorder, as the pathology occurs on the neuronal level and it is accounted by aberrant developmental processes occurring during fetal, childhood, or adolescent periods [1]. The main symptoms of this disorder are represented by disturbance in perception, volition, thinking, fluency, and language production as well as the recognition and expression of emotion that led to impairments in social and occupation functioning [2].

Schizophrenia can be better described by two types of symptoms: negative and positive. Positive symptoms can be more easily understood as the presence of symptoms such as delusions (false fixed beliefs), hallucinations (perceptual disturbances which may occur in any sensory modality but are most commonly auditory), and abnormal psychomotor activity that is usually manifested as grossly disorganized behavior, posturing, and/or catatonia [3]. They are also known as psychotic symptoms, whereas negative symptoms are better at describing the absence characterized by social withdrawal, avolition, alogia, and anhedonia. Besides these two categories, there is a third that is better described as containing the core features of the illness—cognitive symptoms [4].

There are four categories of animal models, which are: neurodevelopmental, pharmacological, lesion, or genetic modeling, but still none of them succeeded to be a typical model in all domains and symptoms of schizophrenia. Developing a more reliable model for this complex disorder is crucial to allow a better understanding of its neurobiological basis [5].

That is why our hypothesis is based on combining more than one model, represented here by ketamine administration as a pharmacological model and social isolation as a neurodevelopmental one.

Ketamine is an FDA-approved anesthetic agent. However, growing evidence indicates that ketamine causes neurotoxicity in developing animal models [6]. Ketamine, as a glutamate antagonist, is well known to induce transient symptoms similar to those observed in patients with schizophrenia [7]. It can produce psychotomimetic effects and also can elicit negative and cognitive symptoms. Such effects have been reported following the acute administration of ketamine as a glutamate antagonist to healthy volunteers [8], and administration of these agents to patients with schizophrenia exacerbates symptoms [9]. Using sub-anesthetic doses of ketamine (5–10 mg kg^−1^) in rodents is commonly used to induce features of schizophrenia-like symptoms [10].

Lahti et al. in 2001 [9] evaluated the mental status of normal and schizophrenic volunteers on the day of ketamine infusion “before then 20, 90, and 180 min after infusion” using the Brief Psychiatric Rating Scale (BPRS) as well as its subscale scores [11]. Authors found a close similarity of mental disorders between ketamine-induced and the schizophrenic volunteers’ own symptoms in addition to their exacerbation with ketamine. Taken together, ketamine is used to induce transient schizophrenia-like symptoms [12]. However, in animal models, ketamine failed to induce chronic schizophrenia-like symptoms on its chronic administration at a dose of 30 mg/kg for 5 consecutive days [13].

Several experimental studies regarding schizophrenia have reported increased reactive oxygen species (ROS) levels and some studies referred this to alterations in antioxidant enzymes, which may have decreased [14], increased [15], or have even remained unchanged [16]. For example, an article from 2003 reported results that demonstrated elevated levels of blood superoxide dismutase (SOD) in neuroleptic-free schizophrenic patients [17]. In a follow-up study by the same authors, a reduction in the SOD levels was reported following risperidone treatment [18]. In addition, higher levels of SOD and glutathione peroxidase (GSH-Px) were present in another experiment, which compared neuroleptic-free schizophrenic patients with schizophrenic individuals who received treatment with haloperidol [19]. However, some other authors reported a statistically significant decrease in levels of SOD, GSH-Px, and catalase (CAT) in schizophrenic individuals who were treated with either atypical or typical antipsychotics [20,21].

The post-weaning social isolation paradigm is used for inducing chronic schizophrenia-like symptoms. It can reproduce some of the behavioral, structural, and neurochemical alterations found in schizophrenic patients [22]. Social isolation (SI) during early postnatal development leads to several long-lasting abnormal behavioral and neuropsychiatric disorders in mice. This was confirmed by subjecting rats to social isolation rearing conditions, at the age of 21 days (just at time of weaning) of postnatal life and for 5 weeks, leading to a series of schizophrenia-related deficits, such as social withdrawal, anxiety disorder, cognitive impairments, sensorimotor gating disturbances, and aggressive behavior [23].

The etiology of schizophrenia is multifactorial: a combination of genetics, brain chemistry, and environment contribute to the development of the disorder. Therefore, we investigated the effect of combined ketamine administration together with social isolation for induction of chronic schizophrenia-like symptoms, aiming to reach a valid animal chronic schizophrenia model that would help to study the disease etiology, pathophysiology, and to investigate emerging treatment.

## 2. Materials and Methods

### 2.1. Experimental Animals

Forty-two male adult Wistar rats of age ranging from 8–10 weeks were used in the current study. The animals were purchased from the animal house of Research Institute of Ophthalmology (RIO) and the study was conducted in the RIO and physiology department, Faculty of Medicine Cairo University. 

Adult male Wistar (*n* = 42) rats, weighing 200–250 g at the beginning of the experiment, were housed in groups of five animals per cage and kept in a room with controlled temperature (22 °C) and a 12:12 h light/dark cycle (starting at 08:00 h), with food and water ad libitum. The animals were treated in accordance with the guidelines of animal bioethics from Cairo University Institutional Animal Care and Use Committee (CU-IACUC) no. CU/III/F/57/18.

### 2.2. Materials and Drugs

Ketamine (Astrapin, Germany), was dissolved in physiological saline, then intraperitoneally injected at a volume of 1 mL/100 g body weight (30 mg/kg).

Aripiprazole (Sigma-Aldrich, St. Louis, MO, USA) was dissolved in 5% Tween 80 in sterile saline and injected intraperitoneally (1 mL/kg, 10 mg/kg).

### 2.3. Experimental Design

#### Pharmacological Treatment

The animals were randomly divided into four main groups. Group 1 (*n* = 7) was used as the control group by administrating saline solution through intraperitoneal injection, whereas Group 2 (*n* = 7) was used to assess social isolation without a pharmaceutical treatment. Group 3 (*n* = 14) was subjected to ketamine injection and then randomly divided into 2 equal subgroups: 3a (*n* = 7) and 3b (*n* = 7), and Group 4 (*n* = 14) was used to illustrate the effects of both ketamine injection and social isolation by randomly splitting the group in two equal subgroups—4a (*n* = 7) and 4b (*n* = 7).

The rats from the control group were housed together in one cage after the saline solution was injected (i.p.); six weeks later, rats underwent behavioral and cognitive evaluation.

From the isolation-only group, each rat was housed individually in a separate cage for a period of three weeks before being housed as a group for another three weeks. Finally, rats underwent behavioral and cognitive evaluation.

The ketamine injection group underwent behavioral and cognitive evaluation immediately after the ketamine injection and brain samples were collected, whereas in the ketamine + group housing group, rats were caged in a group for six weeks before again undergoing behavioral and cognitive evaluation, as well as brain samples collection.

Animals involved in Group 4 were subjected to social isolation for a period of three weeks after the intraperitoneal administration of ketamine injection. After three more weeks, where the rats had been housed together in the same cage, the whole group underwent behavior and cognitive evaluation. The rats from the ketamine + isolation group have been sacrificed and brain samples were collected, while the ketamine + isolation + aripiprazole group started daily treatment of aripiprazole by intraperitoneal means for three weeks. At the end of the treatment period, the ketamine + isolation + aripiprazole group of rats were subjected to behavioral and cognitive studies and brain samples were collected. We added the treatment with a newer atypical antipsychotic, such as aripiprazole, to our sixth group in order to observe if it would reduce the schizophrenia-like symptoms in comparison to the other groups.

Ketamine was administrated at a sub-anesthetic dose (30 mg/kg i.p.), with the observation that it could induce a significant decrease in activity time, reflecting the depressant effect; however, this effect was observed to be acute [13].

Aripiprazole is a second-generation antipsychotic that is commonly used to treat schizophrenia in adult humans. Aripiprazole can reduce both positive and negative symptoms; the chosen dose induces antipsychotic-like effects in rats and mice, which may, therefore, reflect human therapeutically relevant doses [24,25].

On testing days, aripiprazole was given 30 min before starting the behavioral examination and the dosage for aripiprazole was 10 mg/kg/day i.p.

### 2.4. Behavioral Tests

All behavioral tests were conducted in the light period during a specific time frame (9 a.m.–1 p.m.) and each group was tested separately.

#### 2.4.1. Open Field Test

In order to assess anxiety levels, locomotion, and exploration eagerness, we opted to use an open field test.

This apparatus consisted of a square box (80 × 80 × 50 cm), and the field was divided into 25 squares.

The rat’s behavior was recorded for 5 min with a camera placed above the arena measuring the time spent in the central square, the number of crossed lines, as well as grooming and rearing [26].

#### 2.4.2. Elevated Plus Maze Test

Anxiety was measured by using an elevated plus maze test during the behavioral and cognitive studies according to the instructions described in the previous subsection. The maze was built with two open and two closed arms (50 × 10 × 40 cm) mounted 50 cm above the smooth floor.

Rats were placed in the intersection of the four arms of the elevated plus maze and their behavior was recorded for five minutes by using a camera placed above the arena. The recorded behaviors are the time spent in free arms, number of entries made in the open, amount of head dipping, and stretched-attend postures [27].

The order of the tested animals within a group was chosen randomly.

#### 2.4.3. Y-Maze Continuous Procedure

For this behavioral test, the apparatus consists of the Y-maze where the rat is placed in the maze for a defined period (typically 5 min) with the sequence of arm choices being recorded [28].

### 2.5. Biochemical Estimations

After the behavioral and cognitive studies performed at the 6 weeks mark, rats from Group 1 and Group 2 were sacrificed (using Pentobarbital at a dose of 40–50 mg/kg i.p., rats were anesthetized then were killed by decapitation) and the brain removed. Samples were also collected from Group 3a but shortly after the injection and behavioral tests, whereas Group 3b was sacrificed after a 6-week period. In Group 4, brain samples were collected after 6 weeks (with 3 in isolation) in Group 4a and after another 3 weeks for Group 3b.

After homogenization and centrifugation of brain tissues, the supernatant was removed. Then, it was tested for determination of cortical and hippocampal levels of superoxide dismutase, glutathione peroxidase, and malondialdehyde.

#### 2.5.1. Superoxide Dismutase (SOD) Determination

SOD enzyme activity was assessed through the inhibition of nitro blue tetrazolium (NBT) reduction by O_2_—generated by the xanthine/xanthine oxidase system and according to the manufacturer’s instructions. One unit of SOD activity was defined as the amount of enzyme causing 50% inhibition in 1mL reaction solution/mg protein. The values were expressed in U/mg of protein [29].

#### 2.5.2. Glutathione Peroxidase (GPx) Determination

The activity of GPx was evaluated with a GPx detection kit according to the manufacturer’s instructions. GPx catalyzes the oxidation of glutathione by cumene hydroperoxide. In the presence of glutathione reductase and NADPH, the oxidized glutathione immediately converted to the reduced form with a concomitant oxidation of NADPH to NADP+. The decrease in absorbance was measured spectrophotometrically against a blank at 340 nm [30].

#### 2.5.3. Malondialdehyde (MDA) Determination

The MDA level was assayed using routine colorimetric methods. The absorbance of the organic phase was determined at the wavelength of 530 nm. The values are expressed in mmol/mg of protein [31].

### 2.6. Data Analysis

All data obtained through behavioral analysis by the means of four behavioral tests (open field test, elevated plus maze, Y-maze continuous) as well as through biochemical estimations (superoxide dismutase, glutathione peroxidase, malondialdehyde concentration) were coded and entered using the statistical package SPSS version 25. Data were summarized using mean and standard deviation. Comparisons between groups were done using analysis of variance (ANOVA) with multiple comparisons post hoc test in normally distributed quantitative variables, while non-parametric Kruskal–Wallis test and Mann–Whitney test were used for non-normally distributed quantitative variables [27]. *p*-values less than 0.05 were considered as statistically significant.

## 3. Results

### 3.1. Open Field Test

In the open field test, the following parameters were recorded and measured: number of lines, crossings (number of times the line of a square is crossed by the animal with all four legs).

Regarding the number of line crossings, we obtained a significant group difference (F(5,50) = 2265.6 (*p* < 0.001)). The post hoc comparison between the control group and every other group showed significant differences (*p* < 0.001), meaning that the number of line crossings were significantly lower in all groups when compared to the control group. Furthermore, when comparing the social isolation group with all other experimental groups, significant differences were also observed (with *p* lower than 0.001 for all comparisons). When comparing the ketamine group with the ketamine + housing group and with the ketamine + isolation group, statistically significant differences were reported (*p* < 0.001 for both comparisons). The only non-significant difference regarding the number of line crossings in the open field test was noticed when we compared the ketamine group with the ketamine + isolation + aripiprazole (*p* = 0.730). Between the ketamine + housing group and ketamine + isolation and ketamine + isolation + aripiprazole, there were also significant differences (*p* < 0.001). Lastly, a significant difference was also noted between the ketamine + isolation group and the ketamine + isolation + aripiprazole group (*p* < 0.001) (Figure 1). All *p*-values obtained regarding the number of crossings in the open field test can be seen in Table 1.

### 3.2. Elevated Plus Maze

In the elevated plus maze, the following parameters where measured and compared: time spent in open arms (s) and number of arm entries (Figure 2).

In terms of the time spent in open arms (s), we obtained a significant group difference ((F(5,50) = 16.4 (*p* < 0.001). Post hoc comparisons showed no significant difference in terms of time spent in open arms between the control group and each of the following groups: the isolation group (*p* = 1.000), ketamine group (*p* = 0.7462), ketamine + social housing group (*p* = 0.9992), and ketamine + isolation group (*p* = 0.7742). However, the rats in the control group spent significantly less time in the open arms compared with the ketamine + isolation + aripiprazole group (*p* < 0.001). A similar pattern was noted for the isolation group, with no significant difference reported when compared with the ketamine group (*p* = 0.808), ketamine + housing group (*p* = 1.000), and ketamine + isolation group (*p* = 0.833). However, a significant difference was found between the isolation group and the ketamine + isolation + aripiprazole group (*p* < 0.001). In addition, no significant difference was observed between the ketamine group and ketamine + housing group (*p* = 0.874), ketamine and ketamine + isolation group (*p* = 1.000), but a significant difference was noted between the ketamine group and ketamine + isolation + aripiprazole group (*p* < 0.0001). We also observed that there was no significant difference between the ketamine + social housing group and ketamine + isolation group (*p* = 0.894), but a significant difference between ketamine + social housing group and ketamine + isolation + aripiprazole group (*p* < 0.001), whereas between the group of rats which received the ketamine + isolation intervention and the ketamine + isolation + aripiprazole group, there was also a significant difference (*p* < 0.001). Overall, we can say that in terms of the time spent in open arms (s), there was a significant difference between the subgroup treated with ketamine, subjected to social isolation, and given treatment with aripiprazole and all the other groups tested (*p* < 0.001) (Figure 2). All *p*-values obtained regarding the time spent in open arms in the elevated plus maze can be seen in Table 2.

A closer look at the ketamine + isolation + aripiprazole group reveals that rats number 3, 4, and 6 spent the majority of their time in the open arms (297, 300, and 294, respectively), while rat number 2 seemed to be unresponsive to aripiprazole treatment (only 10 s spent in open arms) (Table 3).

When analyzing the results of the behavioral test of the elevated plus maze type, in terms of the number of open arm entries, we obtained a significant group difference (F(5,50) = 48.3 (*p* < 0.001)). Post hoc comparisons showed no significant difference in terms of the number of open arm entries between the control group and the isolation group (*p* = 0.840) and between the control and ketamine group (*p* = 0.345). However, the control rats had a significantly higher number of open arms entries compared to rats in the ketamine + social housing group (*p* = 0.0001), ketamine + isolation group (*p* = 0.001), and ketamine + isolation + aripiprazole group (*p* = 0.042). In addition, we observed a significant difference between the isolation group and ketamine group (*p* = 0.001), ketamine + isolation group (*p* < 0.001), and ketamine + isolation + aripiprazole (*p* = 0.006), with no significant difference between the isolation group and ketamine + social housing group (*p* = 0.757). Other significant differences were noted between the ketamine group and ketamine + social housing group (*p* < 0.001) and ketamine + isolation + aripiprazole (*p* < 0.001), with no significant difference between the ketamine group and ketamine + isolation group (*p* = 0.282). There was also a significant difference between the ketamine + social housing group and ketamine isolation group, as well as with ketamine + isolation + aripiprazole group (*p* < 0.001 for both comparisons) and between ketamine + isolation and ketamine + isolation + aripiprazole (*p* = 0.002) (Figure 3). In addition, all *p*-values obtained in the post hoc analysis regarding the number of open arm entries in the elevated plus maze can be seen in Table 4.

### 3.3. Y-Maze

Regarding to the behavior performance of rats in the Y-maze test, we obtained a significant group difference (F(5,50) = 12.5, (*p* < 0.001)) regarding spontaneous alternation percentage, which might suggest a significant influence on the short-term memory. When the control group was taken for analysis, we noticed that the rats in this group had a significantly higher percentage of spontaneous alternation when compared with the ketamine group (*p* < 0.001) and with ketamine + isolation group (*p* = 0.009). When comparing the group that was subjected only to social isolation, the sole significant difference follows the same pattern with the ketamine group (*p* < 0.001) and ketamine + isolation group (*p* = 0.004). The group that received the ketamine injection exhibited significant differences only when compared to the ketamine + social housing group (*p* < 0.001) and ketamine + isolation + aripiprazole (*p* < 0.001). In terms of the ketamine + social housing group, we observed a significant difference with the ketamine + isolation group (*p* = 0.007), but not with the ketamine + isolation + aripiprazole group. Between the two groups subjected to ketamine injection and social isolation, with the only difference being the aripiprazole treatment for the latter group, a significant difference was also observed (*p* = 0.022) (Figure 4). All *p*-values obtained in the post hoc analysis regarding the percentage of spontaneous alternation in the Y-maze test can be seen in Table 5.

### 3.4. Biochemical Markers

When it comes to the biochemical analysis performed on the brain samples collected from all groups as previously described, we measured and compared three parameters that are considered to be biomarkers for oxidative stress: malondialdehyde (MDA), glutathione peroxidase (GPx), and superoxide-dismutase (SOD), measured both in hippocampus and cortex samples.

Regarding the level of MDA from the hippocampus samples, we found a significant difference between our experimental groups (F(5,50) = 1474.2, (*p* < 0.001)), and the post hoc analysis revealed significant differences (*p* < 0.05) between all compared groups, with one exception between ketamine + social housing group and ketamine + isolation + aripiprazole treatment, where the difference was not significant (*p* = 1.0000) (Figure 5). Therefore, the MDA level in the hippocampus was significantly lower in the control group compared to every other group of rats. However, for the cortex sample, we observed significant group differences for MDA levels between our experimental groups (F(5.50) = 1277.9, *p* < 0.001). Moreover, when we performed post hoc analysis, we observed significant differences between all groups (*p* < 0.05) aside from the isolation group when compared to ketamine + social housing group, illustrating a not significant difference (*p* = 0.0585) (Figure 6). Accordingly, the MDA level in the cortex was significantly lower in the control group compared to every other group of rats. All the *p*-values obtained after the post hoc analysis for the MDA levels for both hippocampus and cortex level can be seen in Table 6 and Table 7.

Concerning the GPx levels from the hippocampus samples, we observed a significant difference between all of the experimental groups (F(5,50) = 1471.3, *p* < 0.001). Post hoc analysis illustrated that all differences were significant between the compared experimental groups (*p* < 0.05) apart from the comparison between the isolation group and the ketamine + isolation + aripiprazole group, which revealed a non-significant difference (*p* = 1.0000) (Figure 7). Thus, the GPx level in the hippocampus was significantly higher in the control group compared to every other group of rats. At the same time, the levels of GPx from the cortex samples revealed a significant group difference (F(5,50) = 210.5, *p* < 0.001). Additionally, we observed by means of post hoc comparisons significant differences (*p* < 0.05) between all groups, except between subgroups, the ketamine + social housing group, and the ketamine + isolation + aripiprazole group (*p* = 1.0000) (Figure 8). Therefore, the GPx level in the cortex was significantly higher in the control group compared to every other group of rats. All *p*-values following the post hoc comparison regarding the GP x level in the hippocampus and cortex can also be seen in Table 8 and Table 9.

In what refers to another oxidative stress marker, the level of SOD from the hippocampus revealed a significant group difference between our experimental groups (F(5,50) = 778.7, *p* < 0.001). Post hoc analysis illustrated significant differences (*p* < 0.05) between all groups compared, with the exception of the comparison between the ketamine + social housing group and ketamine + isolation + aripiprazole group, which presented a non-significant difference (*p* = 1.00). Accordingly, the SOD level in the hippocampus was significantly higher in the control group compared to every other group of rats (Figure 9). When analyzing the results from the cortex SOD samples, we found a significant group difference between all of our experimental groups (F(5,50) = 153.8, *p* < 0.001). Post hoc analysis showed significant differences between all possible comparisons, excluding the comparison between the isolation group and the ketamine + social housing group (*p* = 1.0000); and ketamine + social housing group and the ketamine + isolation + aripiprazole group (*p* = 0.07731), which were both statistically non-significant. Therefore, the GPx level in the cortex was significantly higher in the control group compared to every other group of rats (Figure 10). All the *p*-values obtained after the post hoc analysis regarding the SOD levels can be seen in Table 10 and Table 11.

Interestingly, when we performed the Pearson correlations between the behavioral parameters that we determined in this experiment (number of line crossings in open field test; time spent in open arms and number of open arm entries in elevated plus maze; spontaneous alteration percentage in Y-maze) and the main oxidative markers measured in hippocampus and cortex samples, we obtained some significant correlations. Rats from the isolation group that had a higher percent of spontaneous alternation tended to have significantly higher levels of GPx in the hippocampus (*n* = 7, r = 0.813, *p* = 0.026) and higher levels of cortical MDA (*n* = 7, r = 0.942, *p* = 0.001).

In addition, rats which were treated with ketamine plus social isolation that spent more time in the open arms (s) tended to have significantly higher levels of cortical MDA (*n* = 7, r = 0.589, *p* = 0.027) and higher levels of GPx (cortex) (*n* = 7, r = 0.610, *p* = 0.021).

For the group which was also subjected to aripiprazole on top of ketamine injection and social isolation, we noticed that the rats that spent more time in the open arms tended to have higher levels of cortical MDA (*n* = 7, r = 0.783, *p* = 0.037) and higher levels of GPx in the cortex (*n* = 7, r = 0.818, *p* = 0.025).

For the control group, the ketamine group, and the ketamine + social housing group, no significant correlations were found.

## 4. Discussion

In the present study, we investigated the potential of a combined low dose of ketamine and social isolation in order to establish a better animal model of chronic schizophrenia-like symptoms with the use of male albino rats. The statistical analysis of our data showed higher levels of schizophrenia-like symptoms in the rats which received the isolation plus ketamine experimental intervention when compared to the control group. In addition, the same significant increase in the schizophrenia-like symptoms was found when we compared the isolation plus ketamine group with the ketamine plus social housing group, suggesting that the social isolation alone may play an important role in our model of schizophrenia. This finding is in concordance with the available data, which show that social isolation increases schizophrenia-like symptoms when compared to socially raised rats in non-pharmacological animal models [32,33]. However, to our knowledge, it is the first time that social isolation has been used in addition to ketamine and antipsychotic intervention. In our animal model, in the ketamine plus social isolation group, the number of line crossings in the open field test, the number of open arm entries in the elevated plus maze, and the spontaneous alternations percentage in the Y-maze were significantly lower in the ketamine + isolation group compared to both control and ketamine + social housing group (*p* < 0.05).

Furthermore, the ketamine + isolation intervention significantly increased the MDA levels and decreased the GPx levels both in the hippocampus and the cortex of the rats when compared to both the control and the ketamine + social housing group. The role of social isolation in increasing various oxidative stress markers have been studied before and demonstrated. The authors of an animal model of Alzheimer’s indicated a significant elevation in brain MDA content to 188% as compared to the corresponding control socialized group [34].

In addition, our hypothesis, according to which the aripiprazole treatment will significantly reduce the negative effects of ketamine intervention plus social isolation, was confirmed. When we compared the group of rats which received the ketamine plus the social isolation with the group that received the aripiprazole treatment on top of the ketamine plus social isolation, our statistical analysis revealed significant ameliorations of schizophrenia like-symptoms in the majority of our behavioral tests. Specifically, the antipsychotic treatment with aripiprazole significantly improved the number of line crossings in the open field test, the time spent in open arms, and the number of open arm entries in the elevated plus maze and the percentage of spontaneous alternations in the Y-maze. All these results may be proof that combining ketamine with social isolation could create an animal model capable of illustrating both negative and positive symptoms of schizophrenia.

The use of ketamine as a pharmacological product is widespread due to its multiple medical and clinical usages including analgosedation, anesthesia, pain therapy, and neuroanesthesia. In time, it has also become a good option for creating a pharmacological model for schizophrenia, as it induces psychosis-like effects by binding to the N-methyl-D-aspartate receptor (NMDAR) (non-competitive NMDA antagonist) [35,36,37].

The three types of schizophrenia symptoms (positive, negative, and cognitive) help the disorder to be divided into four categories: premorbid phase (social deficits), prodromal phase (positive symptoms), psychotic phase (florid positive symptoms), and stable phase (negative symptoms) [38,39].

Due to the high complexity of this disorder and the difficulty of treating negative and cognitive symptoms of schizophrenia (the current established therapy instead focuses on the positive symptoms), which have a major influence on a patient’s life quality and the way he is integrated into society, there is an urgent need of an animal model which can mimic the deficits in these symptoms. Currently, there is a variety of model animals for schizophrenia, as well as for other neurodevelopmental disorders, ranging from pharmacological models to genetic or lesion models. However, the available animal models replicate only the positive symptoms with very few models for the negative ones, including learning and memory impairment. This only highlights the need for a novel animal model that approaches these deficits that are resistant to the current antipsychotic therapy [5].

Ellison hypothesized in 1995 [40] that ketamine represents a weaker candidate for a possible schizophrenia model, but Becker demonstrated the opposite by inducing behavioral changes in rats injected with a dose of 30 mg/kg ketamine for 5 consecutive days [13]. Another study that opted for using the same dose for the same period of time revealed that ketamine enhanced immobility in a forced swimming test, which might represent negative symptoms for schizophrenia [41]. Rushford and his group used ketamine at a dose of 10/30 mg/kg for 5 days to induce odor span task performance deficits with a persistence of 21 days [42]. Some mimicking deficits related to cognition (flexibility) were observed when ketamine induced attention deficits in attentional set shifting task in rats [43]. Interestingly, the study of Keilhoff reported an increased neurogenesis in the hippocampus of the rats treated with subanesthetic doses of ketamine [44].

Another study conducted by Ke Xu evaluated the dimensions of positive and negative syndrome scale (PANSS) between the ketamine users and schizophrenia patients with the conclusion that there is a similarity between the two types of psychosis (ketamine and schizophrenia) [45]. One study in particular chose to use repeated subanesthetic doses of ketamine (30 mg/kg) for 5 consecutive days in order to provide evidence for the usefulness of ketamine for obtaining a legitimate animal model for schizophrenia and they reported significant changes in nitrergic and GABAergic systems similar to those observed in the schizophrenic human brain, but they also observed an increase in the density of reduced nicotinamide adenine dinucleotide phosphate diaphorase, neuronal nitric oxide synthase, and cFOS-positive hippocampal interneurons [46]. Another group of researchers that used a subanesthetic dose of ketamine demonstrated significant effects such as impaired memory, with lasting impact on the encoding of sensory stimuli, which comes in support of the idea that repeated doses of ketamine might have a harmful effect on brain function, specifically on the information processing [47]. A study using schizophrenic volunteers has illustrated that the administration of subanesthetic doses of ketamine (0.1, 0.3, 0.5 mg/kg) increase psychosis and positive symptoms [48].

The use of acute doses of ketamine for creating an animal model has demonstrated the appearance of cognitive deficits and hyperlocomotion with long-term memory being improved by therapeutic GTN (glyceryl trinitrate) and SNP (single nucleotide polymorphisms) treatment [49]. It is well known that schizophrenic patients suffer from neuropsychologic impairments, and a study conducted by Moser suggested that some of these impairments seen in schizophrenia are linked with symptoms of depression [50] as suggested by the study of Kalai, which revealed that the severity of a psychotic episode in acute phase schizophrenia predicts the severity of concurrent depressive and anxiety features [51].

A study conducted on 10 psychiatrically healthy individuals which received high doses of ketamine revealed that ketamine induced changes in recall and recognition memory, verbal fluency and nystagmus, and other abnormalities characteristic for schizophrenia [52]. In a one-year longitudinal study conducted by Morgan, the effects of chronic ketamine (long-term effects) were investigated, with the results revealing cognitive deficits and decreased performance of spatial working memory and pattern recognition memory tasks, as well as dissociative symptoms and increased depression scores, suggesting that ketamine has harmful effects on cognitive function and psychological wellbeing [53].

Ketamine and amphetamine have been validated to induce stereotyped behavior in humans [54]. Compared to amphetamine, which induces only the positive symptoms of schizophrenia, subanesthetic doses of ketamine have been demonstrated to produce positive and negative symptoms with those seen in this disorder [55]. The behavioral animal models we used in our experiment have been utilized extensively to describe the effects of both amphetamine and ketamine in the animals [56]. In addition, it is already demonstrated that ketamine is an appropriate drug to induce psychotic behavior, yet there is a debate in the literature regarding the optimal dose. There are studies indicating that ketamine in low doses (for example 5, 10, or 15 mg/kg) can be used as an anti-depressive drug [57]. To induce schizophrenia-like symptoms, it is used in ranges from 10 to 50 mg/Kg [58,59] and, in even higher doses, ketamine is a demonstrated potent anesthetic agent [60].

Oxidative stress is described by an imbalance between reactive oxygen species (ROS) and antioxidants. In schizophrenia patients, it has been reported that there are an increased production of ROS and decreased levels of antioxidant protection, which suggest a potential role of oxidative stress in the etiology of schizophrenia [61]. Most studies performed on schizophrenia patients reported a significant decrease in GPx-specific activity and a significant increase of MDA levels and SOD-specific activity when compared to the control group [62]. However, some investigators found decreased SOD activity in patients with schizophrenia [63]. Therefore, drug-free patients with a first episode of schizophrenia show an increased level of SOD in some studies [64] but also decreased activity in others [65]. This can be explained by a reduced mean duration of illness (4.46 days in the mentioned study) [65]. These low levels of SOD activity might also indicate a compromised antioxidant defense at the onset of schizophrenia. Therefore, the decrease in SOD activity observed in our study might be explained by the duration of the disorder. It is expected that with the progression of the illness, the SOD activity will increase as a compensatory reaction to oxidative stress.

Also, it was suggested that oxidative stress is the main agent responsible for parvalbumin inhibitory interneurons deficit, which is a linker to schizophrenia [66].

As we described before, negative symptoms of schizophrenia are a challenge to mimic in an animal model. Our model and protocol are unique as they combine social isolation and ketamine administration in order to provide an authentic model for negative and positive symptoms. Our results have not only illustrated behavioral changes, but also proof of oxidative stress in the brain samples by the elevated levels of MDA that we observed. The search for effective and valid animal models for both the negative and positive symptoms of schizophrenia has not yet been very successful. The classical approach, examining the effects of various drugs given to animals to induce psychosis, has generally failed to induce both negative and positive symptoms. By combining the effect of ketamine with social isolation, we created a potentially very valuable animal model. In addition, aripiprazole administration has been found to reduce some of the negative symptoms of schizophrenia. The efficiency of this antipsychotic on reducing some negative symptoms in our rats’ sample of schizophrenia brings additional evidence that a model of schizophrenia with both negative and positive symptoms could be developed.

## 5. Conclusions

In summary, we proposed and demonstrated the potential of using our protocol in order to attain a good animal model of schizophrenia capable of illustrating both positive and negative symptoms by combining ketamine with social isolation. Our premise of creating a model capable of exhibiting both positive and negative symptoms of schizophrenia was also based on adding the aripiprazole treatment to a group of rats. Therefore, we compared the ketamine + social isolation group with the ketamine + social isolation + aripiprazole group in order to attempt to discover if the antipsychotic drug would significantly decrease the potential positive schizophrenia-like symptoms induced by social isolation and ketamine. Given that we obtained significant results, we cautiously presume that this might be an important step in developing our animal model capable of illustrating both positive and negative of schizophrenia. Our study has several limitations, such as the fact that we did not also follow the route of a repeated administration of ketamine and the sample sizes are rather small (seven individuals per group); a follow-up study might be indicated.

## Figures and Tables

**Figure 1 brainsci-11-00917-f001:**
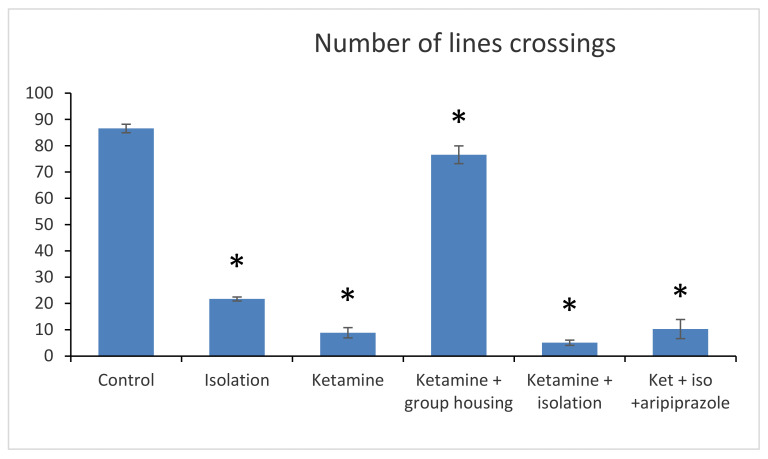
Behavioral results in the open field test of the six analyzed groups based on the number of line crossings. The values are mean. Group 1—control, Group 2—social isolation, Subgroup 3a—ketamine (30 min after injection), Subgroup 3b—ketamine (end of study), Subgroup 4a—isolation + ketamine (3 weeks after isolation), Subgroup 4b—isolation + ketamine + aripiprazole treatment (6 weeks from the starting point). * Significantly different when compared to control group (*p* < 0.05).

**Figure 2 brainsci-11-00917-f002:**
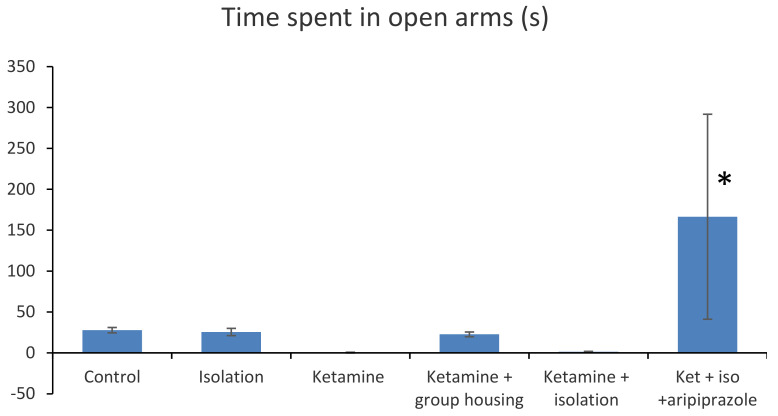
Behavioral results in the elevated plus maze of the six analyzed groups based on the time spent in open arms (s). The values are mean. * Significantly different when compared to control group (*p* < 0.05).

**Figure 3 brainsci-11-00917-f003:**
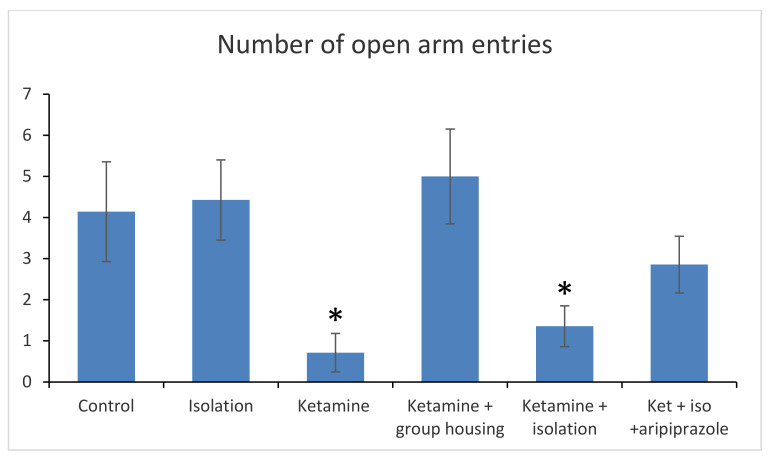
Behavioral results in the elevated plus maze of the six analyzed groups based on the number of open arms entries. The values are mean. * Significantly different when compared to control group (*p* < 0.05).

**Figure 4 brainsci-11-00917-f004:**
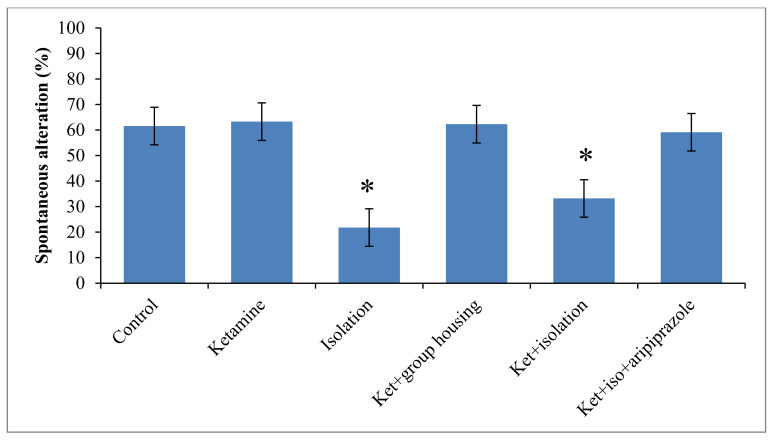
The effects of social isolation, ketamine, social isolation + ketamine, and social isolation + ketamine + aripiprazole on the spontaneous alternation percentage in the Y-maze test. The values are mean ± SD (*n* = 7 for each group); 1 vs. 3a (*p* = 0.00005), 1 vs. 4a (*p* = 0.0094). * Significantly different when compared to control group (*p* < 0.05).

**Figure 5 brainsci-11-00917-f005:**
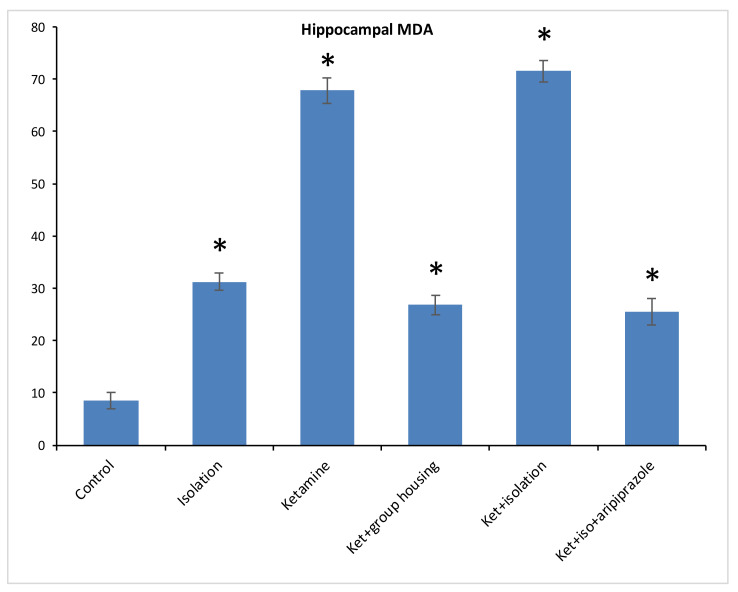
The effects of social isolation, ketamine, combined ketamine and social isolation, and combined ketamine, social isolation, and aripiprazole treatment on MDA-specific activity in hippocampus brain samples (measured in mmol/mg of protein). The values are mean ± SD (*n* = 7 per group). * Significantly different when compared to control group (*p* < 0.05).

**Figure 6 brainsci-11-00917-f006:**
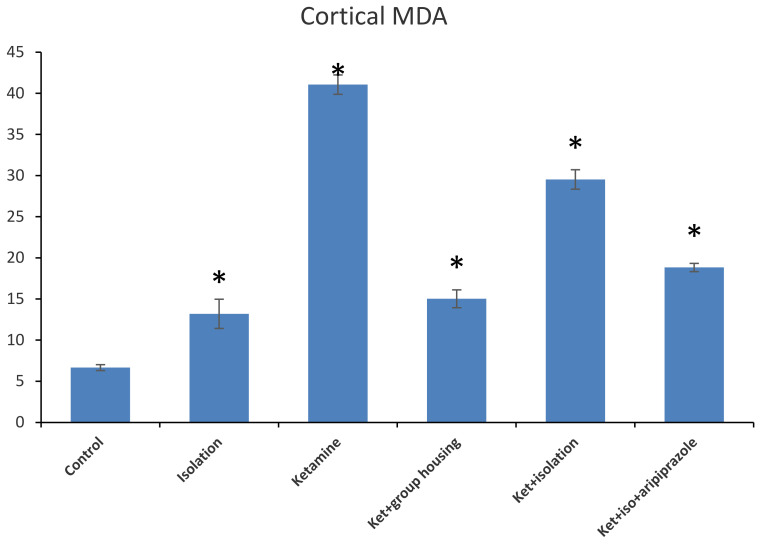
The effects of social isolation, ketamine, combined ketamine and social isolation, and combined ketamine, social isolation, and aripiprazole treatment on MDA-specific activity in cortex brain samples (measured in mmol/mg of protein). The values are mean ± SD (*n* = 7 per group). * Significantly different when compared to control group (*p* < 0.05).

**Figure 7 brainsci-11-00917-f007:**
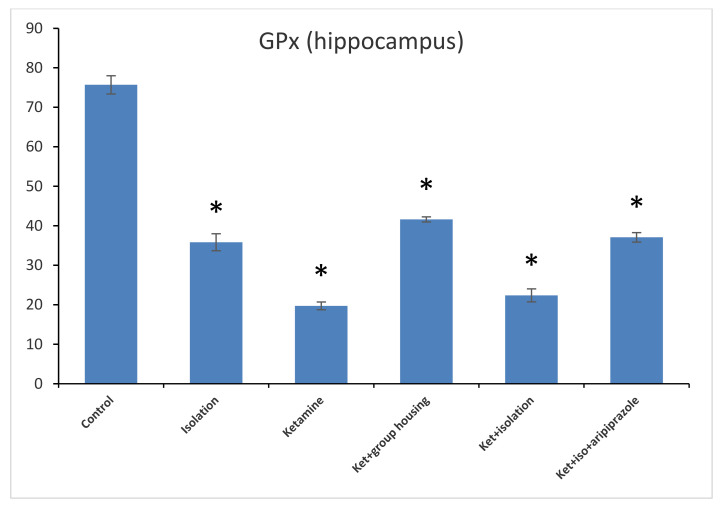
The effects of social isolation, ketamine, combined ketamine and social isolation, and combined ketamine, social isolation, and aripiprazole treatment on GPx-specific activity in hippocampus brain samples (measured in nmol/mg of protein). The values are mean ± SD (*n* = 7 per group). * Significantly different when compared to control group (*p* < 0.05).

**Figure 8 brainsci-11-00917-f008:**
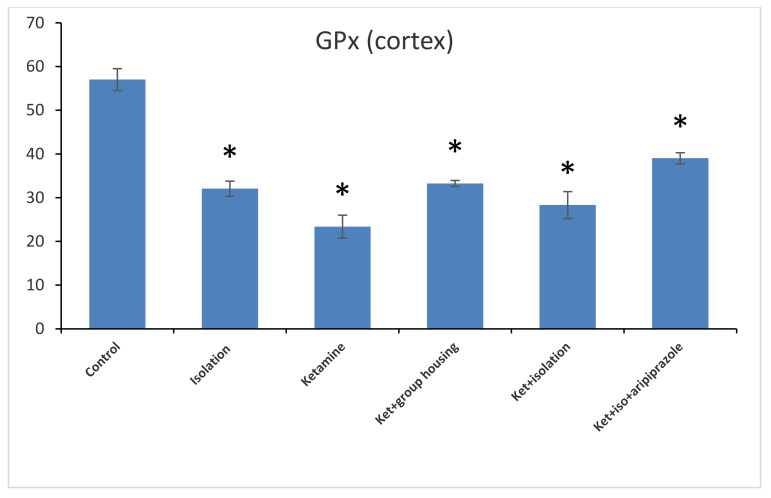
The effects of social isolation, ketamine, combined ketamine and social isolation, and combined ketamine, social isolation, and aripiprazole treatment on GPx-specific activity in cortex brain samples (measured in nmol/mg of protein). The values are mean ± SD (*n* = 7 per group). * significantly different when compared to control group (*p* < 0.05).

**Figure 9 brainsci-11-00917-f009:**
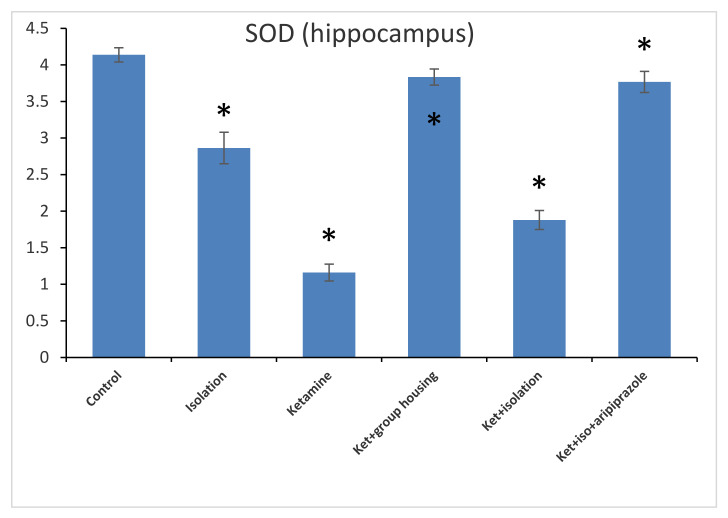
The effects of social isolation, ketamine, combined ketamine and social isolation, and combined ketamine, social isolation, and aripiprazole treatment on SOD-specific activity in hippocampus brain samples (measured in U/mg of protein). The values are mean ± SD (*n* = 7 per group). * Significantly different when compared to control group (*p* < 0.05).

**Figure 10 brainsci-11-00917-f010:**
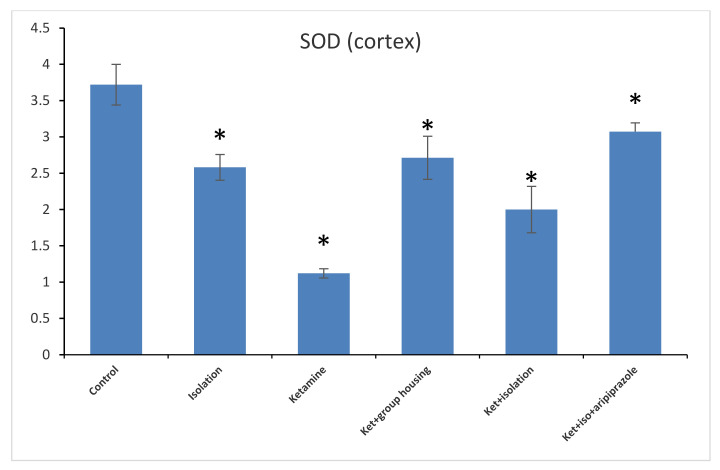
The effects of social isolation, ketamine, combined ketamine and social isolation, and combined ketamine, social isolation, and aripiprazole treatment on SOD-specific activity in cortex brain samples (measured in U/mg of protein). The values are mean ± SD (*n* = 7 per group). * Significantly different when compared to control group (*p* < 0.05).

**Table 1 brainsci-11-00917-t001:** All *p*-values for the number of line crossings in the open field test.

	Control	Isolation	Ketamine	Ketamine + Group Housing	Ketamine + Isolation	Ketamine + Isolation + Aripiprazole
Control	1	<0.001	<0.001	<0.001	<0.001	<0.001
Isolation		1	<0.001	<0.001	<0.001	<0.001
Ketamine			1	<0.001	<0.001	0.730
Ketamine + group housing				1	<0.001	<0.001
Ketamine + Isolation					1	<0.001
Ketamine + isolation + aripiprazole						1

**Table 2 brainsci-11-00917-t002:** All *p*-values for the time spent in open arms in the elevated plus maze.

	Control	Isolation	Ketamine	Ketamine + Group Housing	Ketamine + Isolation	Ketamine + Isolation + Aripiprazole
Control	1	1	0.746	0.999	0.774	<0.001
Isolation		1	0.808	1	0.833	<0.001
Ketamine			1	0.874	1	<0.001
Ketamine + group housing				1	0.894	<0.001
Ketamine + Isolation					1	<0.001
Ketamine + isolation + aripiprazole						1

**Table 3 brainsci-11-00917-t003:** The time spent in open arms by each individual rat from the ketamine + isolation + aripiprazole group.

Rat Number	1	2	3	4	5	6	7
Time spent in open arms (s)	80	10	297	300	294	88	96

**Table 4 brainsci-11-00917-t004:** All *p*-values for the number of open arms entries in the elevated plus maze.

	Control	Isolation	Ketamine	Ketamine + Group Housing	Ketamine + Isolation	Ketamine + Isolation + Aripiprazole
Control	1	0.840	<0.001	0.345	<0.001	0.042
Isolation		1	<0.001	0.757	<0.001	0.006
Ketamine			1	<0.001	0.282	<0.001
Ketamine + group housing				1	<0.001	<0.001
Ketamine + Isolation					1	0.002
Ketamine + isolation + aripiprazole						1

**Table 5 brainsci-11-00917-t005:** All *p*-values for the percentage of spontaneous alternation in the Y-maze test.

	Control	Isolation	Ketamine	Ketamine + Group Housing	Ketamine + Isolation	Ketamine + Isolation + Aripiprazole
Control	1	0.982	<0.001	0.982	<0.001	0.822
Isolation		1	<0.001	1.000	0.004	0.914
Ketamine			1	<0.001	0.611	<0.001
Ketamine + group housing				1	0.007	0.914
Ketamine + Isolation					1	0.022
Ketamine + isolation + aripiprazole						1

**Table 6 brainsci-11-00917-t006:** All *p*-values for the level of MDA in the hippocampus.

	Control	Isolation	Ketamine	Ketamine + Group Housing	Ketamine + Isolation	Ketamine + Isolation + Aripiprazole
Control	1	<0.001	<0.001	<0.001	<0.001	<0.001
Isolation		1	<0.001	0.003	<0.001	<0.001
Ketamine			1	<0.001	<0.001	<0.001
Ketamine + group housing				1	0.545	<0.001
Ketamine + Isolation					1	<0.001
Ketamine + isolation + aripiprazole						1

**Table 7 brainsci-11-00917-t007:** All *p*-values for the level of MDA in the cortex.

	Control	Isolation	Ketamine	Ketamine + Group Housing	Ketamine + Isolation	Ketamine + Isolation + Aripiprazole
Control	1	<0.001	<0.001	<0.001	<0.001	<0.001
Isolation		1	<0.001	0.058	<0.001	<0.001
Ketamine			1	<0.001	<0.001	<0.001
Ketamine + group housing				1	<0.001	<0.001
Ketamine + Isolation					1	<0.001
Ketamine + isolation + aripiprazole						1

**Table 8 brainsci-11-00917-t008:** All *p*-values following the post hoc comparison for the level of GPx in the hippocampus.

	Control	Isolation	Ketamine	Ketamine + Group Housing	Ketamine + Isolation	Ketamine + Isolation + Aripiprazole
Control	1	<0.001	<0.001	<0.001	<0.001	<0.001
Isolation		1	<0.001	<0.001	<0.001	1
Ketamine			1	<0.001	<0.001	<0.001
Ketamine + group housing				1	<0.001	<0.001
Ketamine + Isolation					1	<0.001
Ketamine + isolation + aripiprazole						1

**Table 9 brainsci-11-00917-t009:** All *p*-values following the post hoc comparison for the level of GPx in the cortex.

	Control	Isolation	Ketamine	Ketamine + Group Housing	Ketamine + Isolation	Ketamine + Isolation + Aripiprazole
Control	1	<0.001	<0.001	<0.001	<0.001	<0.001
Isolation		1	<0.001	<0.001	0.018	<0.001
Ketamine			1	<0.001	<0.001	<0.001
Ketamine + group housing				1	<0.001	1
Ketamine + Isolation					1	<0.001
Ketamine + isolation + aripiprazole						1

**Table 10 brainsci-11-00917-t010:** All *p*-values following the post hoc comparison for the level of SOD in the hippocampus.

	Control	Isolation	Ketamine	Ketamine + Group Housing	Ketamine + Isolation	Ketamine + Isolation + Aripiprazole
Control	1	<0.001	<0.001	0.001	<0.001	<0.001
Isolation		1	<0.001	<0.001	<0.001	<0.001
Ketamine			1	<0.001	<0.001	<0.001
Ketamine + group housing				1	<0.001	1
Ketamine + Isolation					1	<0.001
Ketamine + isolation + aripiprazole						1

**Table 11 brainsci-11-00917-t011:** All *p*-values following the post hoc comparison for the level of SOD in the cortex.

	Control	Isolation	Ketamine	Ketamine + Group Housing	Ketamine + Isolation	Ketamine + Isolation + Aripiprazole
Control	1	<0.001	<0.001	<0.001	<0.001	<0.001
Isolation		1	<0.001	1	<0.001	0.003
Ketamine			1	<0.001	<0.001	<0.001
Ketamine + group housing				1	<0.001	0.077
Ketamine + Isolation					1	<0.001
Ketamine + isolation + aripiprazole						1

## Data Availability

All data is available on request from the authors.

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
