# Peer review of "Combined Low Dose of Ketamine and Social Isolation: A Possible Model of Induced Chronic Schizophrenia-Like Symptoms in Male Albino Rats"

_brainsci, 2021, doi:10.3390/brainsci11070917_

Round 1
Reviewer 1 Report
I have a major concern about the authors' claim that their study demonstrated the potential of using their protocol to attain a good animal model of schizophrenia capable of illustrating both positive and negative symptoms by combining ketamine with social isolation. On what basis the author claims their protocol provides positive and negative symptoms are unclear. Moreover, the discussion is redundant of the introduction part.
Author Response
Reviewer #1:
I have a major concern about the authors' claim that their study demonstrated the potential of using their protocol to attain a good animal model of schizophrenia capable of illustrating both positive and negative symptoms by combining ketamine with social isolation. On what basis the author claims their protocol provides positive and negative symptoms are unclear. Moreover, the discussion is redundant of the introduction part.
We improved every section of our article following the first revision. Our entire results section was rewritten. In addition, we improved our whole discussion section and emphasize that future studies are needed to validate our possible model. Also, we already planned our next follow up study with the valuable suggestion received in the present revision.
Reviewer 2 Report
The study illustrates how the combination of low doses of ketamine with social isolation could determine chronic schizophrenia-like symptoms and affect the levels of reactive oxygen species in male albino rats. Some behavioral and biochemical changes were observed, determining a possible first animal model showing chronic schizophrenia-like symptoms.
The main strength of this study lies in the fact that no animal model depicting the positive and the negative symptoms of schizophrenia does exist; up to now, it is possible to study only models representing positive symptoms. Limitations are the small sample involved for each of the groups, the lack of a group of subjects receiving repeated ketamine administrations and the lack of a follow-up.
I found the study interesting, with a great amount of data. Some major issues were nevertheless present in the paper, especially regarding the language (a good check would be useful) and the results section, with some graphs that would need a revision, with maybe some tables to better display the acquired results.
Introduction:
I think that the authors should better develop the role of reactive oxygen species (ROS) alterations (line 77-79) which were observed in schizophrenic patients in previous studies. In this way, also the obtained results would be clearer for the reader.
Overall, the section would benefit of some more references.
Methods:
I found the sections regarding experimental animals, materials and drugs and experimental design quite clear, even if I would develop further the advantages of having one of the subgroups treated with Aripiprazole after ketamine administration.
Regarding the heading “Evaluation of memory function”, I do not think it represents well the aims of all the tests explained in the subsections. At line 157, I would add an explanation of what the “number of crossed lines” is, to better characterize it before using it in the results section. At line 172, “placed horizontally” needs to be clarified.
In the section dealing with the biochemical estimations, a reference should be inserted for the paragraph regarding glutathione peroxidase (lines 197-203).
Results:
This section needs a major revision, being unorganized and quite confusing. Too much numerical results and p-values are presented throughout the main text. Those may be better represented through some tables, that were not used in the paper. In this way, also the differences among the various groups would be properly displayed.
At line 228 “a significant increase in all pairings” needs to be clarified. Moreover, the data present in the same line do not fit with the graph in figure 2.
Across all the results section, the use of “increase between group X and group Y” is not clear.
Line 282: The number of arm entries is the total one or is just related to the closed arms?
A table would be useful to depict the biochemical results.
Discussion
I think that in this section the author explained thoroughly the aim of the study and the novelty that this work would bring. At lines 530-531 “an increased density of reduced nicotinamide adenine dinucleotide phosphate diaphorase” is not clear. Also, at 535-536 the sentence is not complete.
Figures:
I think that modifying the first 2 figures to make them like the rest of the graphs would be better, since the latter ones are more effective at depicting data.
Author Response
Reviewer #2:
Introduction:
I think that the authors should better develop the role of reactive oxygen species (ROS) alterations (line 77-79) which were observed in schizophrenic patients in previous studies. In this way, also the obtained results would be clearer for the reader.
Overall, the section would benefit of some more references.
We added one new paragraph developing the possible involvement of ROS in the pathophysiology of schizophrenia. We also added 5 new references to this section.
Methods:
I found the sections regarding experimental animals, materials and drugs and experimental design quite clear, even if I would develop further the advantages of having one of the subgroups treated with Aripiprazole after ketamine administration.
We explained the main advantage of using aripiprazole in one of our experimental group as the reviewer kindly suggested.
Regarding the heading “Evaluation of memory function”, I do not think it represents well the aims of all the tests explained in the subsections.
We corrected it by renaming the subsection to Behavioral tests
At line 157, I would add an explanation of what the “number of crossed lines” is, to better characterize it before using it in the results section.
We clarified the number of crossed lines as number of times the line of a square is crossed with all 4 legs.
At line 172, “placed horizontally” needs to be clarified.
We corrected it by using “perpendicularly (not facing any of the two arms)”.
In the section dealing with the biochemical estimations, a reference should be inserted for the paragraph regarding glutathione peroxidase (lines 197-203).
We added the missing reference regarding the glutathione peroxidase determination.
Results:
This section needs a major revision, being unorganized and quite confusing. Too much numerical results and p-values are presented throughout the main text. Those may be better represented through some tables, that were not used in the paper. In this way, also the differences among the various groups would be properly displayed.
We reconstructed our entire results section. We added tables with p values for all our variables.
At line 228 “a significant increase in all pairings” needs to be clarified. Moreover, the data present in the same line do not fit with the graph in figure 2.
We corrected the mistakes by reconstructing our entire results section.
Across all the results section, the use of “increase between group X and group Y” is not clear.
We corrected the mistakes by reconstructing our entire results section.
Line 282: The number of arm entries is the total one or is just related to the closed arms?
We corrected the mistake by clarifying we measured the Open arms entries
A table would be useful to depict the biochemical results.
Tables for all our biochemical results were added as required by the reviewer.
Discussion
I think that in this section the author explained thoroughly the aim of the study and the novelty that this work would bring. At lines 530-531 “an increased density of reduced nicotinamide adenine dinucleotide phosphate diaphorase” is not clear.
We rephrased this section.
Also, at 535-536 the sentence is not complete.
We corrected the mistake by rephrasing the sentence.
Figures:
I think that modifying the first 2 figures to make them like the rest of the graphs would be better, since the latter ones are more effective at depicting data.
We remade the first two graphs with the same format as the others as suggested by the reviewer.
Reviewer 3 Report
Summary:
Estephan and colleagues performed a series of experiments with combinations of ketamine and social isolation and found a variety of interesting things. However, these results are buried within a bunch of unimportant information, and they are not grounded by an original hypothesis. The results, as they are currently presented, do not connect to the author’s claim that they are modeling positive and negative symptoms. These connections must be made explicit.
There are three interesting and novel findings in this study. (1) Recent ketamine administration, or ketamine + isolation seems to impair T-maze alternation and spontaneous Y-maze alternation. This could indicate a cognitive impairment, and authors should emphasize this connection. (2) Recent ketamine administration, or ketamine + isolation seems to increase MDA levels in the hippocampus and cortex, indicating oxidative stress. Ariprazole reduces this marker of oxidative stress, although not completely. This was also mirrored by the levels of GPx recorded in these areas. (3) Rats given ketamine and isolation and aripiprazole seem to spend a lot more time in the open arms of the elevated plus maze than all other groups of rats. However, this result seems like it is driven by one or at the most two rats.
The paper also includes findings that replicate other work. This includes the fact that one dose of ketamine, followed by group housing does not interfere with normal fear processes compared to controls. However, social isolation with or without ketamine causes fear-like behaviors in the open field test.
Finally, the paper includes findings that contradict previous findings, or need more follow up. This includes the fact that recent ketamine administration, or ketamine + isolation seems to decrease SOD levels in the hippocampus and cortex. Previous literature examining schizophrenia patients treated with antipsychotics showed elevated SOD levels compared to healthy controls (Padurariu et al. 2010). This must be examined in the discussion. The authors of the current study also correlated many of the behavioral findings in this study showed some correlation to the levels of MDA, GPx or SOD, but failed to account for multiple comparisons, because there were 6 different measures of oxidative stress (3 markers and 2 locations) to use in comparisons.
Methodological questions:
Why did the authors use 30mg/kg of ketamine, when 5-10mg/kg is much more common? Please explain. This dose should make the rats sluggish, as was seen by the lack of movement in the open field test.
Why did the authors use 250g rats? The authors argue that there model helps examine the etiology of schizophrenia, and discuss developmental models. If so, they should have used younger rats.
Why did the authors include grooming, stretching, head dipping & defecation at all? The authors never explain a plausible relationship between these behaviors and schizophrenia.
Why is this a model of SCZ, and not just a model of kids who receive ketamine in the ER? The authors need to build a stronger case for both ketamine and social isolation as being separate environmental stressors if they are trying to build a model of schizophrenia.
Why reactive oxygen species? The authors need to develop the idea that ROS may be part of the etiology of schizophrenia in the introduction and connect it to their hypothesis.
Overall stylistic improvements:
Give the groups names that fit on your figure so that the reader does not have to keep going back to the original figure or methods. Names like:
Group 1 = control
Group 2 = isolation only
Group 3a = ketamine + 30 minutes, sacrifice
Group 3b = ketamine + group housing
Group 4a = ketamine + isolation
Group 4b = ketamine + isolation + aripiprazole
Put significance bars on your figures.
You describe way too many comparisons. Why are you comparing group 3 (combined?) With group 4b in the T-maze test? If you don’t need to make the comparison to make a point, don’t include it.
Scientific accuracy criticisms:
Introduction: Clarify the model in this sentence: “Yet, ketamine failed to induce chronic schizophrenia-like symptoms on its chronic administration at a dose of 30 mg/kg for 5 consecutive days.” (This was in rats, not humans, and you were just talking about humans!)
Materials and drugs: How many mg of ketamine was dissolved in 1 mL? Please give the dosage in mg/kg. This is done later, but put it with the intro to the drugs.
2.3: Was ketamine injected more than once? The figures and text do not imply this. If not, what is the meaning of “after the fifth injection, there was no significant difference in activity time between Ketamine group and saline injected group.”
2.5: Groups 4a and 4b are mixed up.
Results: (page 6 line 246): “we observed a significant increase (p>0.0001) in all pairings.” The way you describe the comparison, it should be significant decrease if your figure is correct. Check all of the times you say ‘increase and decrease.’ In English, we compare the second group mentioned to the first (so if group 2 has more rearing than group 4, a comparison between group 2 and 4 shows a decrease).
Make subheadings for each different test. Open field should be 3.1, elevated plus maze should be 3.2 etc.
Figure 2: What is driving up the mean in subgroup 4b? It looks like one rat spent most of the time in an open arm. Can you expand the y-axis for the other measures and include the time spent in the open arms as an inset?
Use the text to explain why group 4b was so different from all the others in the elevated plus maze. Did one rat spend all his time on the open arms, or did all 7 behave differently from the others? Describe the breakdown of these behaviors in the results section so it is more believable. If this was a consistent result, did you try giving a normal rat the same amounts of aripiprazole? Maybe it makes them especially bold.
Discussion: What is GIN and SNP treatment? Don’t use abbreviations before they are introduced.
Line 57: ROS… suggest a potential role of oxidative stress in the etiology of Schizophrenia, not a certain role.
The authors claim that schizophrenia patients have high MDA and SOD and low GPX in another study. They need to explain why they might have observed the opposite results in their own SOD levels.
Overall, the authors must make connections to show how their behavioral measures are related to positive, negative, or cognitive symptoms. Otherwise, the behavior should not be included. Similarly, they need to develop the idea that this model would be a developmental and neurochemical model of schizophrenia if they want to encourage others to use the model. How is this any improvement over the existing models of schizophrenia?
Author Response
Reviewer #3:
Estephan and colleagues performed a series of experiments with combinations of ketamine and social isolation and found a variety of interesting things. However, these results are buried within a bunch of unimportant information, and they are not grounded by an original hypothesis. The results, as they are currently presented, do not connect to the author’s claim that they are modeling positive and negative symptoms. These connections must be made explicit.
There are three interesting and novel findings in this study. (1) Recent ketamine administration, or ketamine + isolation seems to impair T-maze alternation and spontaneous Y-maze alternation. This could indicate a cognitive impairment, and authors should emphasize this connection. (2) Recent ketamine administration, or ketamine + isolation seems to increase MDA levels in the hippocampus and cortex, indicating oxidative stress. Ariprazole reduces this marker of oxidative stress, although not completely. This was also mirrored by the levels of GPx recorded in these areas. (3) Rats given ketamine and isolation and aripiprazole seem to spend a lot more time in the open arms of the elevated plus maze than all other groups of rats. However, this result seems like it is driven by one or at the most two rats.
The paper also includes findings that replicate other work. This includes the fact that one dose of ketamine, followed by group housing does not interfere with normal fear processes compared to controls. However, social isolation with or without ketamine causes fear-like behaviors in the open field test.
Finally, the paper includes findings that contradict previous findings, or need more follow up. This includes the fact that recent ketamine administration, or ketamine + isolation seems to decrease SOD levels in the hippocampus and cortex. Previous literature examining schizophrenia patients treated with antipsychotics showed elevated SOD levels compared to healthy controls (Padurariu et al. 2010). This must be examined in the discussion. The authors of the current study also correlated many of the behavioral findings in this study showed some correlation to the levels of MDA, GPx or SOD, but failed to account for multiple comparisons, because there were 6 different measures of oxidative stress (3 markers and 2 locations) to use in comparisons.
Methodological questions:
Why did the authors use 30mg/kg of ketamine, when 5-10mg/kg is much more common? Please explain. This dose should make the rats sluggish, as was seen by the lack of movement in the open field test.
We used a higher dose of ketamine because we hypothesized that the effects of ketamine on our behavioral tests depend strongly on the stress state of our rats (isolation versus group housing). For example, in other animal models, higher doses (30 mg/kg) of ketamine produced depression-like behavior in unstressed animals, whereas when administered animals subjected to chronic stress it reduced depression-like behavior [1]. This was in our initial experimental design idea. However, we intend to use different doses of ketamine in our already in progress follow up experiment for better group comparisons (5-10 mg/kg versus 30 mg/kg).
[1]. Fitzgerald PJ, Yen JY, Watson BO. Stress-sensitive antidepressant-like effects of ketamine in the mouse forced swim test. PLoS One. 2019;14(4):e0215554.
Why did the authors use 250g rats? The authors argue that there model helps examine the etiology of schizophrenia, and discuss developmental models. If so, they should have used younger rats.
The reason we used adult rats was that we believed that they were more suited for our ketamine induced part of our model of schizophrenia. However, at the suggestion of the reviewer, we plan to use younger rats in one of our planned follow up study.
Why did the authors include grooming, stretching, head dipping & defecation at all? The authors never explain a plausible relationship between these behaviors and schizophrenia.
We eliminated these variables from our article, leaving only the relevant ones.
Why is this a model of SCZ, and not just a model of kids who receive ketamine in the ER? The authors need to build a stronger case for both ketamine and social isolation as being separate environmental stressors if they are trying to build a model of schizophrenia.
We added a new paragraph in the discussion section in which we explained the advantages of using a ketamine model of schizophrenia.
Why reactive oxygen species? The authors need to develop the idea that ROS may be part of the etiology of schizophrenia in the introduction and connect it to their hypothesis.
We added one new paragraph developing the possible involvement of ROS in the pathophysiology of schizophrenia to the introduction section. We also added 5 new references to this section.
Overall stylistic improvements:
Give the groups names that fit on your figure so that the reader does not have to keep going back to the original figure or methods. Names like:
Group 1 = control
Group 2 = isolation only
Group 3a = ketamine + 30 minutes, sacrifice
Group 3b = ketamine + group housing
Group 4a = ketamine + isolation
Group 4b = ketamine + isolation + aripiprazole
We renamed our experimental groups as requested by the reviewer for an easier understanding of our experimental design and results
Put significance bars on your figures.
We made tables with all p values for all of our relevant variables.
You describe way too many comparisons. Why are you comparing group 3 (combined?) With group 4b in the T-maze test? If you don’t need to make the comparison to make a point, don’t include it.
Our entire results section was remade.
Scientific accuracy criticisms:
Introduction: Clarify the model in this sentence: “Yet, ketamine failed to induce chronic schizophrenia-like symptoms on its chronic administration at a dose of 30 mg/kg for 5 consecutive days.” (This was in rats, not humans, and you were just talking about humans!)
We corrected the mistake.
Materials and drugs: How many mg of ketamine was dissolved in 1 mL? Please give the dosage in mg/kg. This is done later, but put it with the intro to the drugs.
We added the mg/kg dose to the material and drugs section also, as requested.
2.3: Was ketamine injected more than once? The figures and text do not imply this. If not, what is the meaning of “after the fifth injection, there was no significant difference in activity time between Ketamine group and saline injected group.”
We corrected the mistake. The fifth injection referred to the Becker study, not ours.
2.5: Groups 4a and 4b are mixed up.
Results: (page 6 line 246): “we observed a significant increase (p>0.0001) in all pairings.” The way you describe the comparison, it should be significant decrease if your figure is correct. Check all of the times you say ‘increase and decrease.’ In English, we compare the second group mentioned to the first (so if group 2 has more rearing than group 4, a comparison between group 2 and 4 shows a decrease).
Our entire results section was remade and corrected
Make subheadings for each different test. Open field should be 3.1, elevated plus maze should be 3.2 etc.
We made subheadings for each test as suggested by the reviewer.
Figure 2: What is driving up the mean in subgroup 4b? It looks like one rat spent most of the time in an open arm. Can you expand the y-axis for the other measures and include the time spent in the open arms as an inset?
Use the text to explain why group 4b was so different from all the others in the elevated plus maze. Did one rat spend all his time on the open arms, or did all 7 behave differently from the others? Describe the breakdown of these behaviors in the results section so it is more believable. If this was a consistent result, did you try giving a normal rat the same amounts of aripiprazole? Maybe it makes them especially bold.
The mean was not driven up by one rat. We had three rats that spent most of the time in an open arm (300, 297, 294 seconds). However, we did have one rat that seem unresponsive to aripiprazole (only 10 seconds spent in open arms). To clarify, a graph with the exact values of time spent in an open arm of each rat was added to our article.
In addition, at the suggestion of the reviewer, we plan to create a group of rats that will solely receive the aripiprazole treatment for a better isolation of the effect of this drug in our follow up study.
Discussion: What is GIN and SNP treatment? Don’t use abbreviations before they are introduced.
We corrected the mistake and also added explanations for the abbreviations.
Line 57: ROS… suggest a potential role of oxidative stress in the etiology of Schizophrenia, not a certain role.
We corrected it as suggested.
The authors claim that schizophrenia patients have high MDA and SOD and low GPX in another study. They need to explain why they might have observed the opposite results in their own SOD levels.
We added a new paragraph in the discussions section regarding the possible mechanisms that might explain the observed decrease SOD activity in our animal model.
Overall, the authors must make connections to show how their behavioral measures are related to positive, negative, or cognitive symptoms. Otherwise, the behavior should not be included.
We eliminated the irrelevant behavioral variables from our article, leaving only the relevant ones.
Similarly, they need to develop the idea that this model would be a developmental and neurochemical model of schizophrenia if they want to encourage others to use the model. How is this any improvement over the existing models of schizophrenia?
We tried to improve our discussion section and our overall article with the required corrections and additions in order to encourage other authors to use and further validate our model.
Round 2
Reviewer 1 Report
- Although the author substantially improved some parts of the manuscript, still my major concern needs to be addressed. Schizophrenia is characterized mainly by delusion and hallucination, so on what basis the author claims that their animal model of schizophrenia capable of illustrating positive symptoms? I would suggest the author rephrase this terminology.
- Still, the author needs to improve the discussion parts. There is a repetition of the information provided in the introduction section. See line 500-508 and 44-53.
Author Response
Reviewer #1:
- Although the author substantially improved some parts of the manuscript, still my major concern needs to be addressed. Schizophrenia is characterized mainly by delusion and hallucination, so on what basis the author claims that their animal model of schizophrenia capable of illustrating positive symptoms? I would suggest the author rephrase this terminology.
Thank you! Although our model is far from perfect and still in development and we understand the difficulty of illustrating delusion and hallucination in an animal model, our premise of creating a model capable of exhibiting positive symptoms of schizophrenia was based on adding the aripiprazole treatment to a group of rats. Therefore, we compared the ketamine + social isolation group with the ketamine + social isolation + aripiprazole group in order to attempt to discover if the antipsychotic drug will significantly decrease the potential positive schizophrenia like symptoms induced by social isolation and ketamine. Given that we obtained significant results we cautiously presume that this might be an important step in developing our animal model capable of illustrating both positive and negative of schizophrenia.
We added a new paragraph in the discussion section discussing this part of our results. We also rephrased some of our terminology in order to highlight that this is a in- development model and our results are still in the “might” and “potential” stage of work.
- Still, the author needs to improve the discussion parts. There is a repetition of the information provided in the introduction section. See line 500-508 and 44-53.
We improved our discussion section by adding several paragraphs discussing our results. We also removed the repeated information from this section.
Reviewer 3 Report
Overall, the manuscript has improved a great deal. However, there are still some major textual revisions that I would like to see:
1) Weave your own results into the discussion. Right now, you inadequately summarize your own results at beginning of the discussion, then you launch into other similar studies. Make explicit connections between your methods and other teams methods. Make explicit connections between the patterns of oxidative stress markers that you saw and other groups’ results. When do they match? When do they differ?
2) While the tables showing the p-values work as an accurate way to present the information, it makes the paper very cumbersome to read, and I’ve honestly never read a paper like that before. Could you please try putting the comparison bars with stars into the figures themselves?
3) Think carefully about your abstract and reword it to highlight your results. You have grammatical mistakes which make it hard to read.
4) What are the units on figures 7-12?
5) Check your statistics on Figures 9, 10, 11 and 12 (GPx and SOD in the hippocampus and cortex). Usually when I get a p-value of 1 for a comparison, something is wrong with the data cleanup or copying into the stats software, although I suppose you have a lot of comparisons, so that may be driving up the p-value.
6) Throughout the results, refer to the MDA as “hippocampal MDA” instead of MDA (hippocampus) and “cortical MDA” instead of MDA (cortical).
7) Please describe your significant differences as increases or decreases from the baseline/control condition.
8) The Y-maze basically repeats the information from the T-maze experiment. Could you turn one into supplemental information? Both of these mazes are meant to test the cognitive abilities of the rats. Remember, this isn’t your dissertation! You are using the figures to make a coherent scientific argument.
9) Figure 3 should be a table. That will be smaller and easier to read.
10) Figure 4 no longer needs such an extensive legend because the groups are named.
11) State the directions of the correlations for the Pearson’s correlations. (Something like: “Rats that spent more time in the open arms tended to have higher levels of GPx in the hippocampus (stats, stats, stats).”
Typos:
Abstract: Check grammar throughout.
Lines 21-22 “would be of big value” -> “would be valuable for studying”
Line 25: emphasizes (check grammar)
Lines 27-28: Reword to make the results clearer
Introduction:
Page 2, line 68: rodents (check grammar)
Results:
Make sure that all “Table X” and “Figure X” are capitalized.
Page 9, line 309. No abbreviations: No-> number
Discussion:
The second sentence needs to be reworded so that you are summarizing the results, not just listing the things that you measured.
Page 22 line 495: “widespread”, also multiple or various, not both.
Page 22 line 511: “the current established therapy INSTEAD FOCUSES on the positive symptoms.”
Page 22 line 526: “of 10/30 mg/kg” (not OR)
Page 23 line 557: the exact doses in ng/ml are not helpful because everything else has been in mg/kg. Can you convert these or remove these doses?
Page 23 line 570: “in” (not ION)
Page 23 line 581: “in of” select one and delete the other
Page 23 line 583: “some investigators FOUND decreased…”
Page 24, line 592: No need to use the abbreviation PVI, just spell it out. Also LINKED not linker in that same sentence.
Page 24, line 595: Unique or new, not both.

Author Response
Reviewer #3:
Overall, the manuscript has improved a great deal. However, there are still some major textual revisions that I would like to see:
1) Weave your own results into the discussion. Right now, you inadequately summarize your own results at beginning of the discussion, then you launch into other similar studies. Make explicit connections between your methods and other teams methods. Make explicit connections between the patterns of oxidative stress markers that you saw and other groups’ results. When do they match? When do they differ?
We improved the discussions section by adding several paragraphs discussing our results.
2) While the tables showing the p-values work as an accurate way to present the information, it makes the paper very cumbersome to read, and I’ve honestly never read a paper like that before. Could you please try putting the comparison bars with stars into the figures themselves?
We added stars in every graph highlighting the significant differences (p<0.05) between the control group and the other groups as requested by the reviewer.
3) Think carefully about your abstract and reword it to highlight your results. You have grammatical mistakes which make it hard to read.
We rewrote our abstract almost entirely highlighting our results as the reviewer kindly suggested.
4) What are the units on figures 7-12?
We added the units in the legends of all the mentioned figures
5) Check your statistics on Figures 9, 10, 11 and 12 (GPx and SOD in the hippocampus and cortex). Usually when I get a p-value of 1 for a comparison, something is wrong with the data cleanup or copying into the stats software, although I suppose you have a lot of comparisons, so that may be driving up the p-value.
We double checked our data, and the p-values are correct. As the reviewer pointed out some of our p values may be drove up by our multiple comparisons.
6) Throughout the results, refer to the MDA as “hippocampal MDA” instead of MDA (hippocampus) and “cortical MDA” instead of MDA (cortical).
We made the suggested correction.
7) Please describe your significant differences as increases or decreases from the baseline/control condition.
We described all of our significant results as higher/lower than control group as requested by the reviewer.
8) The Y-maze basically repeats the information from the T-maze experiment. Could you turn one into supplemental information? Both of these mazes are meant to test the cognitive abilities of the rats. Remember, this isn’t your dissertation! You are using the figures to make a coherent scientific argument.
At the beginning of our experiment, we decided to use both models because the Y maze has the advantage of more natural angles as compared to the 90 degrees angles of the T maze, which might make learning and performance more difficult for our rats, but the T maze tends to be the more sensitive in some specific phenotypes of Alzheimer’s models. However, the results obtained in these 2 models are similar in our experiment as pointed by the reviewer, so we removed the T maze from our article for being redundant.
9) Figure 3 should be a table. That will be smaller and easier to read.
We transformed Figure 3 in a Table as requested by the reviewer.
10) Figure 4 no longer needs such an extensive legend because the groups are named.
We reduced the legend of Figure 4 as suggested by the reviewer.
11) State the directions of the correlations for the Pearson’s correlations. (Something like: “Rats that spent more time in the open arms tended to have higher levels of GPx in the hippocampus (stats, stats, stats).”
We rephrased the Pearson’s correlations section following the example the reviewer suggested and also stated the direction of our correlation.
Typos:
Abstract: Check grammar throughout.
Lines 21-22 “would be of big value” -> “would be valuable for studying”
Line 25: emphasizes (check grammar)
Lines 27-28: Reword to make the results clearer
We corrected all the mistakes in the abstract and also improved this section.
Introduction:
Page 2, line 68: rodents (check grammar)
We corrected the mistake.
Results:
Make sure that all “Table X” and “Figure X” are capitalized.
We made sure that all Table and Figures are capitalized.
Page 9, line 309. No abbreviations: No-> number
We corrected the abbreviation.
Discussion:
The second sentence needs to be reworded so that you are summarizing the results, not just listing the things that you measured.
We correct it. Thank you.
Page 22 line 495: “widespread”, also multiple or various, not both.
We corrected the mistakes.
Page 22 line 511: “the current established therapy INSTEAD FOCUSES on the positive symptoms.”
We corrected the mistake as suggested by the reviewer.
Page 22 line 526: “of 10/30 mg/kg” (not OR)
We corrected the typo.
Page 23 line 557: the exact doses in ng/ml are not helpful because everything else has been in mg/kg. Can you convert these or remove these doses?
We removed the exact doses from the sentence as requested.
Page 23 line 570: “in” (not ION)
We corrected the typo.
Page 23 line 581: “in of” select one and delete the other
We corrected the mistake.
Page 23 line 583: “some investigators FOUND decreased…”
We corrected the mistake pointed by the reviewer.
Page 24, line 592: No need to use the abbreviation PVI, just spell it out. Also LINKED not linker in that same sentence.
We deleted the abbreviation.
Page 24, line 595: Unique or new, not both.
We corrected the mistake.